# Thermal Evaporation Synthesis, Optical and Gas-Sensing Properties of ZnO Nanowires

Pham Hong Thach [1,2] and Tran Van Khai [1,2,*]

1  Faculty of Materials Technology, Ho Chi Minh City University of Technology (HCMUT), 268 Ly Thuong Kiet Street, District 10, Ho Chi Minh City 700000, Vietnam
2  Vietnam National University—Ho Chi Minh City (VNU-HCM), Linh Trung Ward, Thu Duc District, Ho Chi Minh City 700000, Vietnam
*  Correspondence: tvkhai1509@hcmut.edu.vn; Tel.: +84-70-332-7675

**Abstract:** The purpose of this study is to synthesize and explore the relationship between the optical properties and gas-sensing performance of ZnO nanowires (NWs). Well-aligned ZnO nanowire (NW) arrays were synthesized on a silicon substrate using the thermal evaporation method without any catalyst or additive. The structures, surface morphologies, chemical compositions, and optical properties of the products were characterized using X-ray diffraction (XRD), field emission scanning electron microscopy (FESEM) together with energy-dispersive spectroscopy (EDS), high-resolution transmission electron microscopy (HRTEM), X-ray photoelectron spectroscopy (XPS), and photoluminescence (PL) spectroscopy, and their gas-sensing properties for $NO_2$ were examined. The results showed that single-crystalline ZnO NWs with high density grow uniformly and vertically on a Si substrate. The FESEM and TEM images indicate that ZnO NWs have an average diameter of roughly 135–160 nm with an average length of roughly 3.5 μm. The results from XRD confirm that the ZnO NWs have a hexagonal wurtzite structure with high crystalline quality and are highly oriented in the [0001] direction (i.e., along the *c*-axis). The deconvoluted O 1s peak at ~531.6 eV (29.4%) is assigned to the oxygen deficiency, indicating that the ZnO NWs contain very few oxygen vacancies. This observation is further confirmed by the PL analysis, which showed a sharp and high-intensity peak of ultraviolet (UV) emission with a suppressed deep-level (DL) emission (very high: $I_{UV}/I_{DL} > 70$), indicating the excellent crystalline quality and good optical properties of the grown NWs. In addition, the gas-sensing properties of the as-prepared ZnO NWs were investigated. The results indicated that under an operating temperature of 200 °C, the sensor based on ZnO NWs is able to detect the lowest concentration of 1.57 ppm of $NO_2$ gas.

**Keywords:** sensing; ZnO; thermal evaporation; nanowires; semiconductor

## 1. Introduction

Nitrogen oxide ($NO_2$) is a strong oxidizing gas that has become one of the extremely harmful gases that derive from combustion of fossil fuels and car exhaust [1,2]. Exposure to $NO_2$ gas results in serious damage to lung tissues and reduces the fixation of $O_2$ on red blood corpuscles, and the gas also contributes to acid rain as well as generating increased ozone in the lower atmosphere [3]. Nowadays, due to the increase in the numbers of cars, power plants, and factories, the concentration of $NO_2$ gas in the air atmosphere has increased rapidly [4]. According to health and safety guidelines, humans should not be exposed to more than 2.5 ppm of $NO_2$ [5,6]. Therefore, the development of gas sensors for the precise detection of $NO_2$ is of great importance for environmental monitoring and human health protection.

In recent years, metal oxide semiconductors (MOSs) such as $SnO_2$, ZnO, $In_2O_3$, $WO_3$, $Fe_2O_3$, NiO, $Cu_2O$, etc., have attracted tremendous attention in the field of gas sensing because of their low cost, simple fabrication methods, long life, high stability, and superior

sensing performance [7–13]. Among them, zinc oxide (ZnO) is an n-type semiconductor with a wide direct band gap of 3.37 eV (at 300 K), possessing a large exciton binding energy (60 meV) and high electron mobility (100–200 $cm^2 V^{-1} s^{-1}$) [14,15], making it suitable not only for gas sensing [16] but also for potential applications in solar cells [17], electronic and optoelectronic devices [18], and catalysts [19]. In fact, due to its advantages of high sensitivity, good oxidation resistance, great chemical–thermal stability, environmental friendliness, and low cost, nanosized ZnO with different morphologies has been widely employed to detect various gases including $H_2$, $H_2S$, $NO_2$, $NH_3$, $CH_4$, CO, ethanol, and acetone [16].

It is well known that gas-sensing performance strongly depends upon structural parameters including crystal size, specific surface area, microstructure, crystallographic planes and the crystallinity of sensing materials [20]. For instance, a decrease in crystal size, an increase in surface-to-volume ratio, and high crystallinity are required to enhance the sensitivity of gas sensors. Since sensing reactions take place mainly on the surface of the sensitive material [21–24], the control of the size of the semiconductor materials is one of the first requirements for enhancing the sensitivity of the sensor. Apart from these factors, morphology also plays a critical role in governing sensing performance [25–29]. Therefore, many efforts have been focused on the synthesis of ZnO nanostructures with different morphologies, such as nanoparticles [30], nanorods [31], nanowires [32], nanotubes [33], nanofibers [34], nanoflowers [35], and hierarchical nanostructures [36]. These nanostructured ZnO materials have higher surface-to-volume ratios compared with the bulk material, which leads to there being more active sites for gas absorption and facilitates charge transfer, thereby improving sensing performance. Recently, numerous efforts have been made to investigate the relationship between the morphology and sensing properties of ZnO nanostructures [37–41]. For example, Agarwal et al. [35] synthesized two types of ZnO nanostructures, ZnO nanorods and flower-like ZnO nanostructures, using a hydrothermal method. They reported that compared with nanorods, nanoflowers have a greater surface area and more surface defects. Due to these differences, flower-like ZnO is able to adsorb more target gas molecules, resulting in an enhanced gas response. Several researchers have studied the use of ZnO nanorods for monitoring $H_2S$ gas, and their findings indicate an improved capacity to detect $H_2S$ over other gases as well as showing an increased response and selectivity through the use of ZnO nanorods compared with bulk ZnO material [42,43]. In order to explore morphology-dependent gas-sensing properties, Zhang et al. [44] prepared ITO materials with various morphologies, including film, nanoparticles, nanorods, and nanowires, using a sputtering method. The results of this study indicated that compared with the other materials, the nanowires possess a larger specific surface area, a greater number of oxygen vacancies, and well-defined electron transport pathways, so their sensing properties are the best. Furthermore, by adjusting the density of nanowires, their sensing performance for ethanol gas was greatly enhanced. Compared with other morphologies of ZnO nanostructures, ZnO NWs have recently drawn a large amount of interest because of their distinctive geometric characteristics, high surface-to-volume ratio, well-defined electron transportation direction, and lower agglomeration tendency. These features are able to enhance the electron flow and affect the reaction between surface-adsorbed oxygen and gas molecules, thereby improving the sensing performance [45–48]. Up to now, there have been many reports on the gas-sensing behavior of ZnO NWs for various gases, which have exhibited excellent sensing performance [49–52]. The majority of these ZnO gas sensors, however, are based on a single NW or tangled NW membrane with no alignment. The vertically well-aligned NW arrays can provide a simple matrix for studying the average effect of assembled NWs. However, so far, to the best of our knowledge, little or no work has been reported on the application of vertical ZnO NWs on the Si substrate for $NO_2$ sensing. Recently, various methods have been employed for the synthesis of ZnO NWs, including the hydrothermal [53], thermal evaporation [54,55], chemical vapor deposition (CVD) [49], physical vapor deposition [56], and metal organic chemical vapor deposition (MOCVD) methods [57,58]. Among these methods, thermal

evaporation is frequently employed due to its ease of operation, environmental friendliness, ease of scaling up, relatively low-cost process, etc. Moreover, with the use of this technique, the as-prepared ZnO NWs show good quality [59,60]. Therefore, the purpose of this study is to synthesize and explore the relationship between the optical properties and gas-sensing performance of ZnO NWs using the thermal evaporation method.

## 2. Materials and Methods

### 2.1. Fabrication of ZnO Nanowires

The well-aligned ZnO NWs were grown via thermal evaporation deposition in a horizontal tube furnace (OTF-1200x-STM, MTI Corp., Richmond, CA, USA) with an inner diameter of 720 mm and a heating zone of 400 mm. Silicon wafers (Si (100)) with thickness of 500 μm (Okemetic Co., Ltd., Tokyo, Japan) and size of 1 cm × 1 cm were used as the substrate for the growth of ZnO NWs. Before the substrates were put inside the chamber for growth, they were chemically etched with $H_3PO_4$ (85%, Merck Chemicals Co., Ltd., Darmstadt, Germany) solution for 60 s to remove the native oxide layer ($SiO_2$), then sonicated with a mixture of acetone, methanol, and deionized water in a sonication bath (Elma Select 60) for 10 min, and finally dried by air. The cleaned Si (1 cm × 1 cm) substrates were placed on top of an alumina boat with a diameter and length of 1.5 and 4 cm, respectively, containing about 0.25 g high-quality metallic zinc powder (75 μm, 99.99%, Sigma-Aldrich, Burlington, MA, USA), and then put at the center of the furnace, where the temperature is highest. The vertical distance between the zinc source and the substrate was about 2.5 mm, with a downstream separation of 15 mm. The temperature inside the furnace was raised from 20 °C to the reaction temperature of 620 °C at a rate of 20 °C per min with Ar (99.99%, Bao Khanh Co. Ltd., Ha Noi, Vietnam) at a flow rate of 350 mL/min The zinc source was thermally vaporized to grow ZnO NWs at atmospheric pressure under Ar at a flow rate of 350 mL/min for 90 min at 620 °C. After the reaction, the equipment was switched off and quartz tube was cooled naturally to room temperature under Ar at a flow rate of 100 mL/min.

### 2.2. Material Characterization

The crystal structure and phase composition of synthesized samples were determined via X-ray diffraction using a Bruker D8 Advanced diffractometer (Rigaku, Tokyo, Japan) with CuKα radiation at λ = 1.54178 Å, operating at 40 kV voltage and 200 mA current. Data were taken for the 2θ range of 25 to 75 degrees with a step of 0.02 degree and a scan step time of 25 s. The surface morphology and compositions of the ZnO NWs were examined using a scanning electron microscope (JEOL JSM-5900 LV SEM, Tokyo, Japan) operated at an accelerating voltage of 20 kV and equipped with an energy-dispersive spectroscopy (EDS) microanalysis. The microstructure of the NWs was observed with a high-resolution electron microscope (HRTEM, JEM-2010, JEOL Ltd., Tokyo, Japan) operated at an accelerating voltage of 200 kV. The elemental components and bonding configuration of the ZnO samples were investigated using an X-ray photoelectron spectroscope (XPS, VG Multilab ESCA 2000 system, East Grinstead UK) with a monochromatized Al K X-ray source (hν = 1486.6 eV). Data analysis was carried out with XPSPEAK41 software using mixed Gaussian–Lorentzian functions after Shirley background correction. The photoluminescence (PL) measurements were performed with a laser Raman spectrometer (HORIBA Jobin Yvon, iHR550) at room temperature, using a He-Cd laser-line with an excitation wavelength of 325 nm and at a laser power of 25 mW. For the sensing measurements, a thin (~100 nm) Au film was deposited on the ZnO NW samples via direct current (DC) magnetron sputtering to form electrodes using an interdigital electrode mask. The electrode pattern consisted of seven Au electrode fingers, each of them being 7 mm long and 0.5 mm wide, with 0.5 mm spacing. The gas-sensing characteristics were determined with a Keithley source-meter model no. 2400 connected to computer. Two mass flow controllers (MFC-3660) were applied to adjust the $NO_2$ gas concentration and carrier gas (dry air). A precise concentration of $NO_2$ was produced via dilution from the standard concentration

of NO$_2$ (0.1% NO$_2$ + 99.9 N$_2$), as shown in Table 1. The concentration of the diluted NO$_2$ was calculated using Equation (1):

$$C_{dulited} = C_o \frac{F1}{F1 + F2} \tag{1}$$

where $F_1$ and $F_2$ are flow rate of NO$_2$ and carrier dry air, respectively, and $C_0$ is 0.1%. Total gas flow (target gas and dry air) was kept constant at 500 ppm. The resistance of the sensor in dry air or in the target gas was measured at 200 °C when a potential difference of 1 V was applied between the Au (~100 nm) electrodes. Sensitivity is one of the important factors in determining gas-sensing performance. Mathematically, sensitivity (S) of the ZnO NW sensor is defined as (R$_g$/R$_a$), where R$_a$ and R$_g$ are the electrical resistances of the sensor in air and target gas, respectively. The response time (t$_{response}$) is defined as the period in which the electrical resistance of the sensor reaches 90% of the response value upon exposure to the target gas, while the recovery time (t$_{recovery}$) is defined as the period in which the electrical resistance of the sensor returns to 10% of the response value after the target gas is removed [61]. The sensing characteristics were investigated at NO$_2$ concentrations of 2–10 pm, operating at temperature of 200 °C under atmospheric pressure.

**Table 1.** Volumetric flow rates of the target gas (NO$_2$), dry air, and standard concentration of NO$_2$ (Co = 0.1%).

| MFC-1 NO$_2$ (sccm) | MFC-2 Dry Air (sccm) | NO$_2$ Concentration in Tube: $C_{dulited}$ (ppm) |
|:---:|:---:|:---:|
| 1 | 499 | 2 |
| 3 | 498 | 6 |
| 5 | 495 | 10 |

## 3. Results and Discussion

The crystalline structure and phase composition of the synthesized samples were examined using powder X-ray diffraction. A typical XRD pattern of the obtained ZnO NWs is shown in Figure 1. From the pattern, it can be seen that all diffraction peaks at 2θ~31.8, 34.6, 36.4, 47.6, 56.7, 62.8, 67.8, and 72.5 can be indexed to the reflection from the $(10\bar{1}0)$, $(0002)$, $(10\bar{1}1)$, $(10\bar{1}2)$, $(11\bar{2}0)$, $(10\bar{1}3)$, $(11\bar{2}2)$, and $(0004)$ crystal planes of the hexagonal wurtzite ZnO structure with the cell constants of a = 3.249 Å and c = 5.206 Å (JCPDS card number: 36-1451, P6$_{3mc}$ space group) [62,63]. The lattice constants of the prepared ZnO NWs can be calculated as a = 3.249 Å and c = 5.185 Å with the c/a ratio = 1.5957. The unit cell volume for deposited ZnO NWs was estimated by using [64]: $V = \frac{\sqrt{3}}{2}a^2c$, where a and c are the lattice constants of the ZnO NWs. Accordingly, the calculated value of the unit cell volume for ZnO NWs is 47.4 Å$^3$. Therefore, the lattice parameters (a = b, c, V) estimated for synthesized NWs are in good agreement with the reported crystallographic data for ZnO [62,63,65]. The strongest peak intensity at the (0002) plane suggests that the ZnO NWs are preferentially oriented in the c-axis direction or in a direction perpendicular to the substrate surface. No traces of zinc, impurities, or substrates are detected from this pattern, indicating that the ZnO NW samples were composed of a pure ZnO hexagonal phase. It is widely known that the hexagonal crystal structure of ZnO is composed of an alternating arrangement of tetrahedrally coordinated Zn$^{2+}$ and O$^{2-}$ ions along the c-axis [66]. The most common crystal faces found on the one-dimensional ZnO nanostructures are polar Zn-terminated (0001) and O-terminated $(000\bar{1})$ as well as nonpolar $(01\bar{1}0)$ and $(2\bar{1}\bar{1}0)$ facets [67]. Therefore, adjusting the growth rate along the different directions can induce anisotropic growth of ZnO. Kinetic studies have shown that incoming precursor species tend to preferentially adsorb on the polar surfaces to decrease the surface energy because the polar surfaces are less thermodynamically stable. Accordingly, the higher surface energy facets grow faster than the others, and they usually have a smaller surface

area. The sequence of the surface energy (E) of the facets in different crystallographic directions of ZnO is commonly $E[0001] > E[10\bar{1}\bar{1}] > E[10\bar{1}0] > E[10\bar{1}1] > E[000\bar{1}]$, which is the same order as the crystal growth velocity, indicating the fastest growth rate in the [0001] direction [68]. Thus, in the case of ZnO NWs, the fastest growth rate is along the c-axis, resulting in the anisotropic growth of nanowires. Meanwhile, ZnO NWs with large areas of nonpolar $(01\bar{1}0)$ and $(2\bar{1}\bar{1}0)$ facets are more energetically favorable [67]. This explanation is supported by the XRD results, in which the (0002) peak intensity is much higher than the others, indicating that ZnO NWs were formed along the c-axis in the [0001] direction (i.e., with predominantly exposed polar (0001) facets).

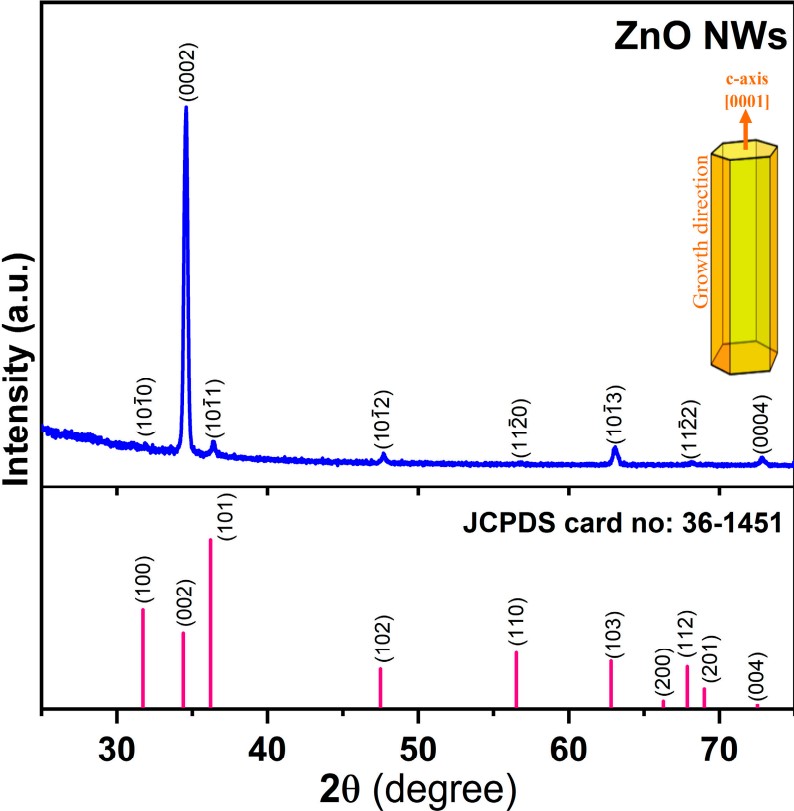

**Figure 1.** XRD pattern of ZnO NWs.

The morphology of the as-prepared samples was investigated using FESEM. Figure 2a–d show typical FESEM images of the obtained ZnO NWs at different magnifications. These sample ZnO NWs were synthesized at a temperature of 620 °C with a synthesis time of 90 min. Figure 2a,b show the low-magnification FESEM images of the deposited ZnO NWs and confirm that large-area, uniform, and dense NWs are vertically grown across the surface of the substrate. The NWs possess a smooth and clean surface along their entire lengths. Figure 2c,d reveal high-magnification images of ZnO NWs, indicating that the top of the NWs is completely flat with a perfect hexagonal structure and the diameter of the NWs is in the range of 160–200 nm with an area density of ~6.85/$\mu$m². Figure 3 illustrates the typical EDS spectrum of the as-grown ZnO NWs, indicating that there is no evidence of metallic elements in the NWs, and the as-prepared ZnO NWs exclusively include zinc and oxygen components. The existence of a Si peak in the spectrum is attributable to the Si substrate.

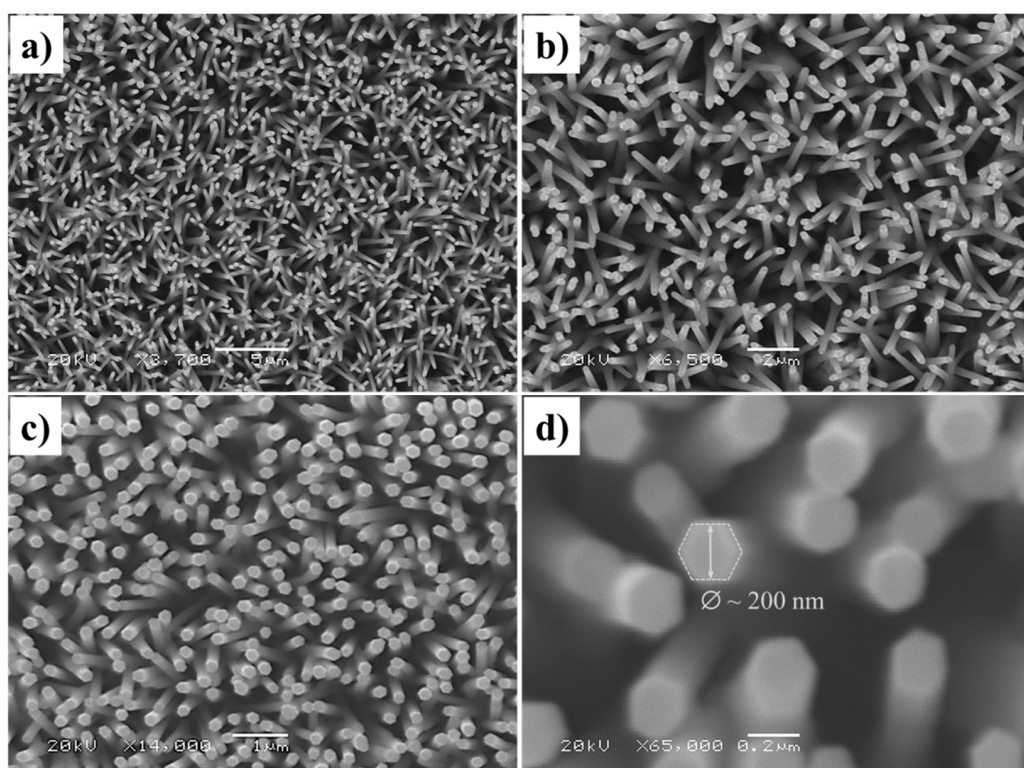

**Figure 2.** FESEM images with different magnifications—(**a**) 3700×; (**b**) 6500×; (**c**) 14,000×; and (**d**) 65,000×—of ZnO NWs.

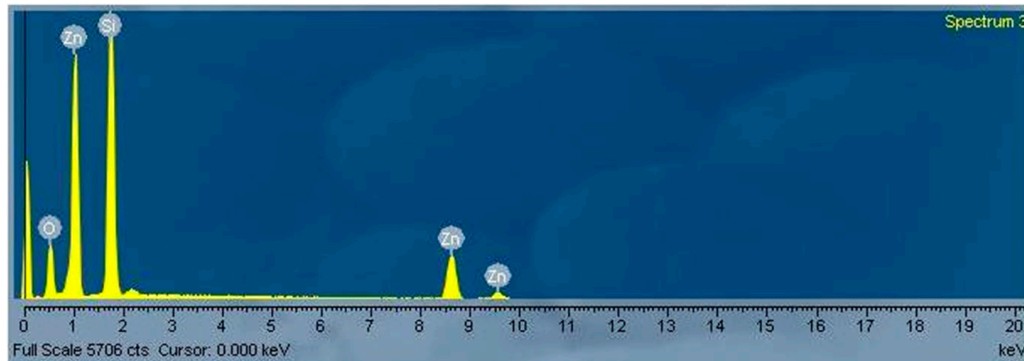

**Figure 3.** The EDS spectrum of ZnO NWs.

Detailed structural features of the as-prepared ZnO NWs are shown in Figure 4a,b. Figure 4a shows that the obtained ZnO NWs possess a very smooth surface and a relatively straight morphology with an average diameter of ~135–160 nm and a length of ~1.5–3.5 μm. A typical HRTEM image is shown in Figure 4b. It can be clearly observed that the ZnO NW contains no defects such as dislocations or stacking faults, indicating that it has a clean surface structure with excellent quality. The inset in Figure 4b is a lattice-resolved HRTEM image of a segment of a single ZnO NW. It can be clearly seen that the ZnO crystal lattice is well oriented and there are no noticeable structural defects throughout the region. This indicates that the obtained ZnO NWs are structurally uniform and defect-free. However, it should not be assumed that the as-synthesized ZnO NW arrays are free of other defects, such as vacancies and gaps, which may be invisible in HRTEM observations. As seen from the inset in Figure 4b, the measured lattice spacing of the ZnO NW was around 0.26 nm, corresponding to the distance between two (0002) crystal planes, indicating that the ZnO NWs are grown in the [0001] direction. This is consistent with the XRD results.

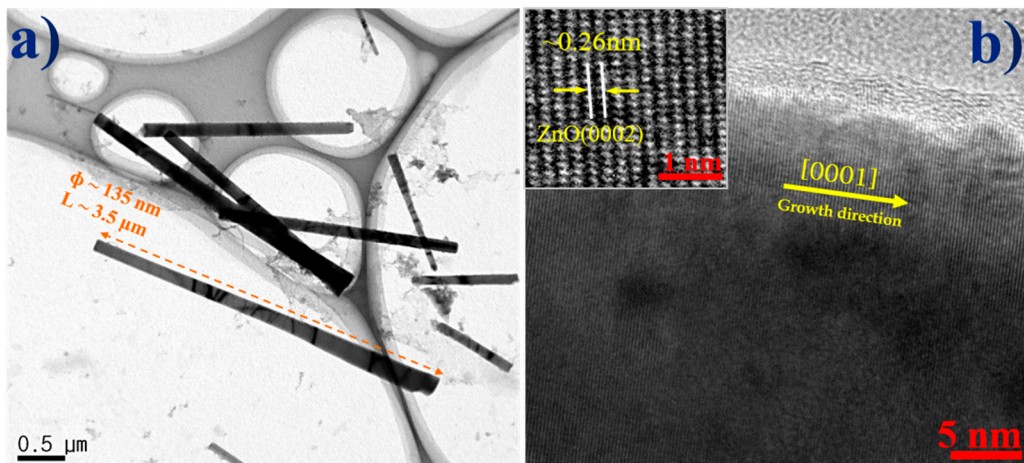

**Figure 4.** TEM images of the synthesized ZnO NWs: (**a**) conventional TEM; (**b**) HRTEM; the inset is a high-magnification TEM image revealing that the measured lattice spacing of 0.26 nm confirms the growth of ZnO NW in the [0001] direction.

The surface chemical compositions and their corresponding valence states of the as-prepared ZnO NWs were further investigated via XPS, and the results are shown in Figure 5a–d. All of the binding energies in the XPS analysis were corrected for specimen charging by reflecting them to the C1s peak position at ~284.6 eV [69]. Figure 5a exhibits the wide-scan XPS spectrum, from which the peaks located at ~284.4, ~533.2, and ~1021.5–1045.5 eV, corresponding to the C1s, O1s, and Zn2p core levels, respectively, can be observed clearly. Here, the peaks for Zn and O are expected to be from the ZnO material, whereas the presence of the C peak is due to the adventitious carbon contamination as well as the chemisorbed $CO_2$ on the surface of samples when exposed to air after the growth, before they are transferred to the XPS chamber (often observed in the XPS spectra of ZnO available in the literature) [70,71]. A more detailed investigation of the chemical states of Zn2p, O1s, and C1s was carried out, taking into consideration the high-resolution scans from core-level XPS, as shown in Figure 5b–d. The XPS spectra were fitted by Gaussian–Lorentzian functions after Shirley background correction using XPSPEAK41 software. Shown in Figure 5b is the high-resolution C1s core-level XPS spectrum, which can be fitted into four components. The main peak is centered at ~284.7 eV and represents C=C bonds due to the adventitious carbon, while the peaks located at ~285.7, ~286.6, and ~288.7 eV can belong to the C-OH group, C-O group, and carbonate phase ($CO_2$), respectively. As recorded, the high-resolution O1s core-level XPS spectrum (Figure 5c) can be deconvoluted into three peaks with binding energies of ~530.3, ~531.6, and ~532.4 eV. The peak located at ~530.3 eV (~24%) is attributed to $O^{2-}$ species in the lattice ($O_{Lat}$) of the wurtzite structure of hexagonal ZnO [72–76], whereas the peak centered at ~531.6 eV (~29.4%) is assigned to the $O^{2-}$ state of oxygen defects such as oxygen vacancies ($V_{\ddot{O}}$), oxygen interstitials ($O_i$), or surface oxygen atoms, suggesting the formation of the nonstoichiometric ZnO. Finally, the peaks at ~532.4 eV (~46.6%) are typically related to the chemisorbed or dissociated ($O_C$) oxygen species on the surface, such as $O_2$, $H_2O$, and $CO_2$ [77–79]. As shown in Figure 5d, the high-resolution Zn2p core-level XPS spectrum can be deconvoluted into two peaks at binding energies of ~1022.4 eV and ~1045.5 eV, which are attributed to the spin–orbits of Zn2p$_{3/2}$ and Zn2p$_{1/2}$ of tetrahedral $Zn^{2+}$, respectively, confirming that the oxidation state of Zn is +2 in the ZnO NWs [80,81]. The binding energy splitting between these two components ($\Delta$BE) is 23.1 eV, which is consistent with previous studies [72].

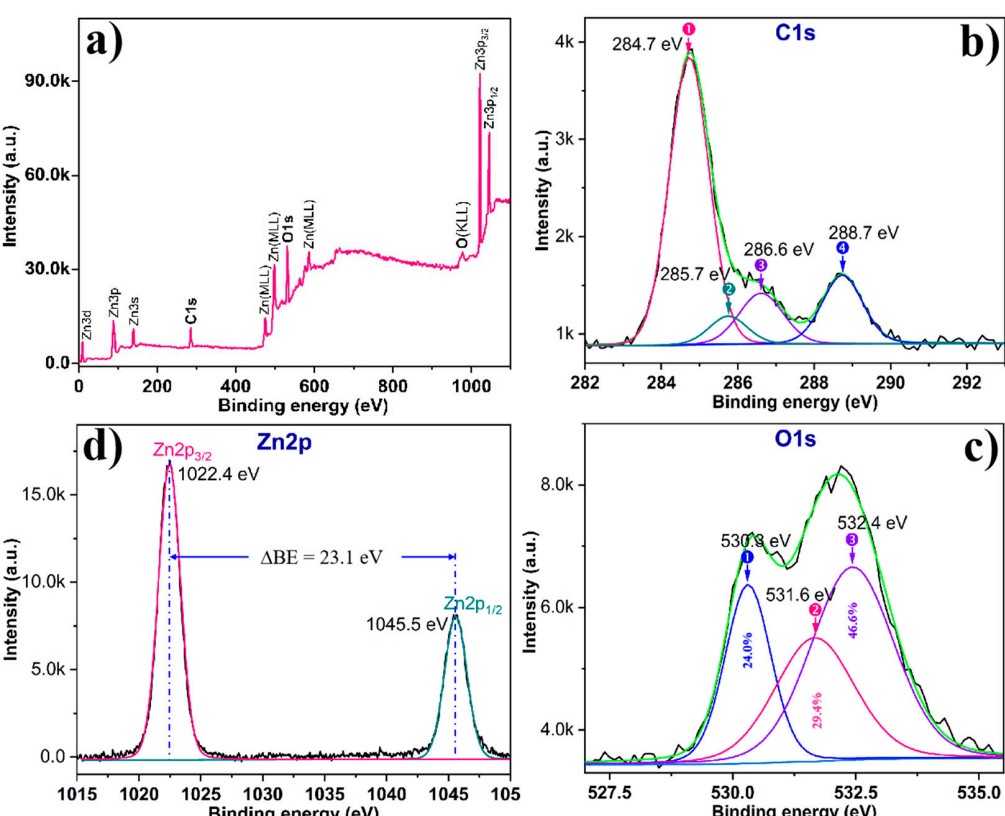

**Figure 5.** (**a**) Wide-scan XPS spectrum and high-resolution scan XPS of (**b**) C1s, (**c**) O1s, and (**d**) Zn2p of ZnO NWs. C1s peak position at ~284.6 eV is used as a binding energy reference to correct the XPS spectra in case of any charging effect.

The room-temperature PL spectrum of the ZnO NWs is presented in Figure 6. The PL spectrum was deconvoluted via Gaussian–Lorentzian fitting. The PL spectrum is very useful for analyzing any possible point defects in the NWs. Figure 6 shows that ZnO NWs have a strong and narrow ultraviolet (UV) emission peak at ~379.5 nm (~3.27 eV) as well as a broad and weak visible emission band that peaks at ~491 nm (~2.53 eV). The UV emission is attributed to the recombination of free excitons between the conduction and valence bands and is called near-band-edge emission (NBE) [82]. On the other hand, the visible emission band is the so-called deep-level (DL) emission of ZnO, which consists of the emissions associated with different types of DL defects in ZnO as well as surface states of ZnO NWs, such as oxygen vacancies ($V_O$) [83,84], zinc interstitials ($Zn_i$) [85], oxygen interstitials ($O_i$), zinc vacancies ($V_{Zn}$), and antisite oxygen ($O_{Zn}$) [86]. Additionally, there is a tiny PL peak at ~390 nm (~3.18 eV), which is caused by the electronic transition from $Zn_i$ states to the valence band maximum [87–89]. The intensity ratio between the UV and DL peaks, $I_{UV}/I_{DL}$, is commonly used to estimate the defect density in ZnO [90]. Among various synthesis techniques, the ZnO NWs grown through thermal evaporation frequently reveal a very intense exciton emission [91–93]. The intensity ratio $I_{UV}/I_{DL}$ recorded from our samples is more than 77, which is higher than those reported in most of the literature (see Table 2) [91–99], indicating that the prepared ZnO NWs have a high crystalline quality. The poor DL emission indicates a low concentration of DL defects in ZnO NWs produced by thermal evaporation.

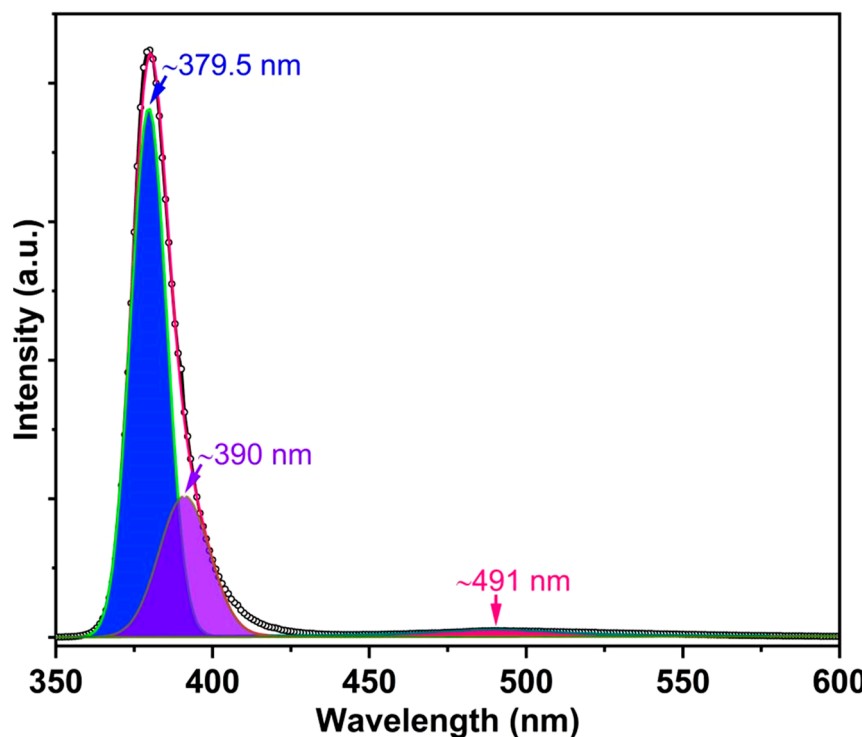

**Figure 6.** Room-temperate PL spectrum of as-prepared ZnO NWs.

**Table 2.** Comparison of intensity ratios IUV/IDL of ZnO nanostructures synthesized using different methods.

| Morphology | Synthesis Method | $I_{UV}/I_{DL}$ | References |
|---|---|---|---|
| ZnO NWs | Pulsed laser deposition | ~11.9–45.4 | [94] |
| ZnO nanorods | Hydrothermal | ~0.3 | [95] |
| ZnO nanorods | Hydrothermal | ~0.6 | [96] |
| ZnO nanorods | Hydrothermal | >19 | [97] |
| ZnO nanorods | Hydrothermal | ~0.6–10 | [98] |
| ZnO nanorods | Thermal decomposition | ~1.8 | [99] |
| ZnO NWs | Physical vapor deposition | 2.5–4.7 | [90] |
| ZnO NWs | Thermal evaporation | ~1.4 | [91] |
| ZnO nanocolumns | Thermal evaporation | ~2.4 | [92] |
| ZnO NWs | Thermal evaporation | ~16.2 | [93] |
| ZnO NWs | Thermal evaporation | >77 | Present work |

The intrinsic point defects in nanostructured ZnO have been demonstrated to play a crucial role in determining the performance of electronic devices [100–103], in terms of doping control, free charge density, minority carrier lifetime, and luminescence efficiency [104]. For instance, it has been shown that by increasing the number of surface defects in ZnO NWs, highly sensitive UV sensors may be produced [102]. These surface defects can serve as trapping or recombination centers for the charge carriers and lead to the formation of a surface depletion region. A greater width of the depletion layer at the NW surface leads to higher UV sensitivity. However, a density of defects on the NW surface that is too high may cause negative effects on nanogenerator devices' performance [100,101]. It has been reported that ZnO nanostructures with high crystalline quality are known to improve photocatalytic activity due to the enhanced separation efficiency of photogenerated electron–hole pairs [105,106]. In the study of nanostructured ZnO-based UV sensors,

high electron mobility is considered to contribute to enhancing the photocurrent [107–109]. High electron mobility may be attained by improving the crystallinity of ZnO. Therefore, precisely controlling the quality of the produced ZnO nanomaterials is critical in the production of a high-performance electronic device. As mentioned above, the optical behavior of the ZnO NWs is studied by analyzing the defect-related deep-level (DL) emission and near-band-edge (NBE) UV emission in the PL spectrum of NWs. Two main features noted in the PL spectrum are located at ~379.5 eV (~3.27 eV) and ~491 nm (~2.53 eV), and respectively labeled as ultraviolet (UV) emission ($I_{UV}$) and deep-level defect emission ($I_{DL}$) peaks. The DL emissions are usually caused by oxygen and zinc vacancies ($V_O$ and $V_{Zn}$) and interstitials ($O_i$ and $Zn_i$), antisites ($O_{Zn}$ and $Zn_O$), and hydrogen impurities [110]. The ratio $I_{UV}/I_{DL}$ can provide a qualitative indicator of the number of radiation defects in the produced nanomaterial. The higher the $I_{UV}/I_{DL}$ ratio, the lower the number of point defects. Therefore, controlling the defect states and entirely suppressing defect-related emissions have become the most important issues for improving UV emission efficiency. DL emission was believed to cause a decrease in UV emission intensity. Many studies have reported that ZnO nanostructures obtained through wet chemical methods [95–98] usually exhibit a low UV emission with relatively strong visible emission ($I_{UV}/I_{DL}$: ~0.3÷19), indicating the presence of a large number of defects in the ZnO nanostructures. This is attributed to the insufficient reaction caused by low synthesis temperature and residual precursors and/or the formation of by-products during the synthesis process. This restricts the application of these ZnO nanostructures in optoelectronic devices as well as biochemical sensors. Unlike those prepared via other methods, the ZnO NWs prepared via thermal evaporation show a very high ratio of $I_{UV}/I_{DL}$, over 77. This confirms the high crystallinity of the as-grown NWs and the existence of a very low number of defects. These properties, along with the lack of grain boundaries and the long-distance order, make ZnO NWs suitable for applications in electronics, optoelectronics, photonics, solar cells, photocatalysis, etc. [111].

Figure 7a indicates typical resistance curves of ZnO NW sensors for 2, 6, and 10 ppm $NO_2$ gas. The $NO_2$ sensing was measured at 200 °C, the moderate working temperature of ZnO-based sensor devices [46,112]. After exposure to $NO_2$ gas, the resistance of the sensor increased, with the n-type behavior of the sensor being derived from the intrinsic n-type behavior of ZnO. The resistance increased upon exposure to $NO_2$ gas and approximately recovered to the initial state after removal of $NO_2$; variations in resistance were found to be reversible behaviors. Also, as predicted, the sensor sensitivities (S) increase when $NO_2$ gas concentrations increase. As seen in Table 3, the sensitivity of ZnO NW sensors changes as a function of $NO_2$ concentration. As the $NO_2$ concentration rises from 2 to 10 ppm, the sensitivity rises from 1.02 to 1.15. Figure 7b displays a typical response and recovery curve of the ZnO NW sensor towards 2 ppm $NO_2$ when operating at 200 °C. When the sensor is exposed to 2 ppm of $NO_2$ gas, an apparent increase in resistance is detected. In this case, the sensor response and recovery times are recorded to be 340 and 760 s, respectively. It was found that $t_{recovery}$ is larger than $t_{response}$; this could be due to the fact that $NO_2$ is strongly bonded to oxygen vacancies on the ZnO NW surface, with almost three times the bond strength of lattice oxygen ($E_{ads} = -0.30$ eV) [113]; as a result, its desorption occurs at a slower rate.

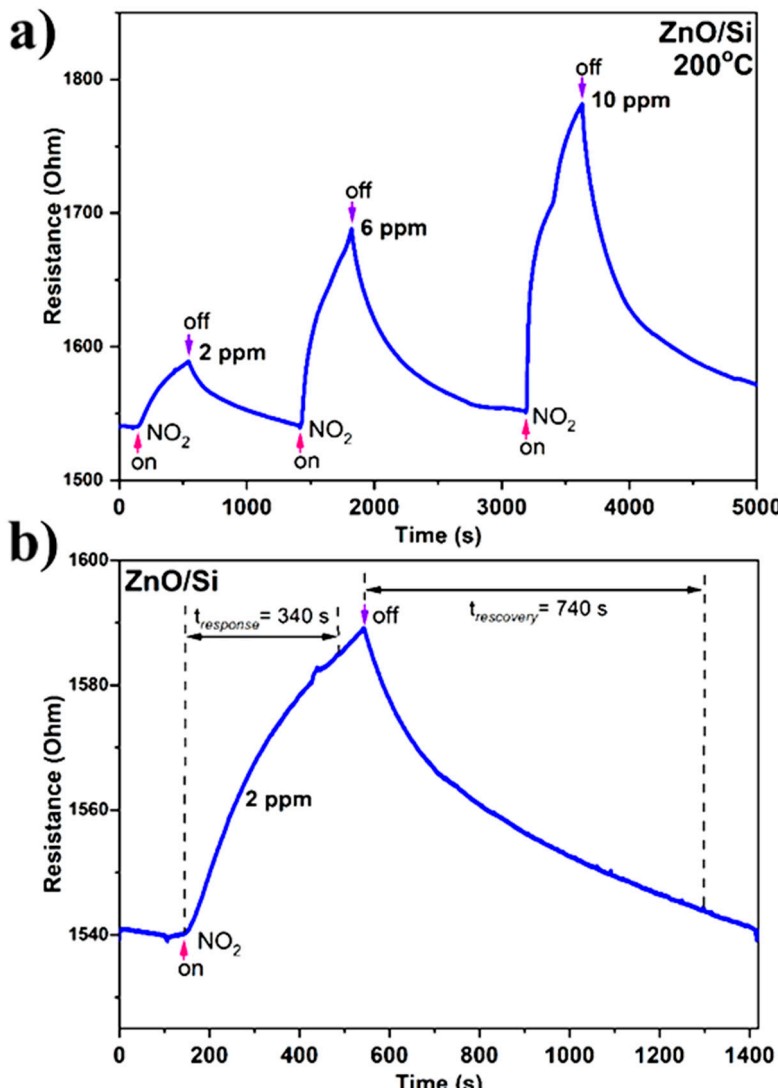

**Figure 7.** (**a**) The changes in the sensor resistance of the as-prepared ZnO NWs in various concentrations of $NO_2$ at operating temperature of 200 °C. (**b**) Typical response and recovery times of the ZnO NW sensor when exposed to 2 ppm $NO_2$ at 200 °C.

**Table 3.** Sensitivities measured at different $NO_2$ concentrations for the ZnO NW sensor at temperature of 200 °C.

| $NO_2$ Conc. (ppm) | Sensitivity (S) |
|---|---|
| 2 | 1.02 |
| 6 | 1.09 |
| 10 | 1.15 |

The $NO_2$ gas-sensing mechanism of ZnO NWs (n-type semiconductor) is mainly based on the electric resistance change when $NO_2$ gases make contact with the surface of the ZnO NWs. In particular, in the air atmosphere, oxygen molecules act as electron acceptors and are chemically adsorbed on the surface of ZnO NWs (see Equation (2)) [114–116]. By capturing electrons from the ZnO conduction band, oxygen molecules ionize on the surface material to form chemisorbed oxygen ions. As a result, an electron-depleted region develops on the ZnO surface, and a potential barrier is formed due to the decrease in electron density, leading to a decrease in the number of electrons in the conduction band of ZnO and an increase in resistance [117]. This is called the baseline resistance in air. The

types of oxygen species adsorbed ($O^{2-}$, $O^{-}$, and $O_2^{-}$) on the ZnO surface depend on the ambient temperature, as can be expressed in Equations (3)–(5) below [118]:

$$O_{2(gas)} \leftrightarrow O_{2(ads)} \tag{2}$$

$$O_{2(ads)} + e^{-}_{(surf)} \leftrightarrow O_2^{-}_{(ads)}, \ T < 100 \ ^{\circ}C \tag{3}$$

$$O_2^{-}_{(ads)} + e^{-}_{(surf)} \leftrightarrow 2O^{-}_{(ads)}, \ 100 \ ^{\circ}C < T < 300 \ ^{\circ}C \tag{4}$$

$$O^{-}_{(ads)} + e^{-}_{(surf)} \leftrightarrow O^{2-}_{(ads)}, \ T > 300 \ ^{\circ}C \tag{5}$$

$$O^{2-}_{(ads)} \leftrightarrow O^{2-}_{(Lat)} \tag{6}$$

At low temperatures, oxygen is commonly adsorbed in its molecular state $O_2^{-}$, whereas at high temperatures it dissociates as atomic $O^{-}$ and $O^{2-}$. Thus, there are two types of oxygen species on the ZnO NW surface; one type is oxygen adsorption and the other is lattice oxygen ($O^{2-}_{(Lat)}$) (see Equation (6)). In this work, the sensors were operated at a low temperature of 200 °C. When the sensor is exposed to the atmosphere of $NO_2$ (strong oxidizing gas), the $NO_2$ molecules not only withdraw the electrons from the ZnO conduction band, but also interact with the oxygen species adsorbed on the ZnO surface because of their stronger electrophilic properties (2.28 eV) compared with oxygen (0.43 eV) [119–121]. This results in the increase in the width of the electron depletion layer and junction potential barriers, which eventually increase the sensor resistance and generate a sensing response [122]. The absorption of $NO_2$ on the ZnO surface is complicated; however, it can be summarized by Equations (7)–(9) [123–127]. When $NO_2$ gas supply is stopped and the sensor is exposed to air again, NO can readily react with dissociated oxygen species and releases the electrons back to the conduction band, causing $NO_2$ to be released into the air and reducing the resistance (see Equations (10)–(13)).

$$NO_{2(gas)} \leftrightarrow NO_{2(ads)} \tag{7}$$

$$NO_{2(ads)} + O_2^{-}_{(ads)} + e^{-}_{(surf)} \leftrightarrow 2O^{-}_{(ads)} + NO_2^{-}_{(ads)}, \ T < 100 \ ^{\circ}C \tag{8}$$

$$NO_2^{-}_{(ads)} + O^{-}_{(ads)} + 2e^{-}_{(surf)} \leftrightarrow 2O^{2-}_{(ads)} + NO_{(gas)}, \ 100 \ ^{\circ}C < T < 300 \ ^{\circ}C \tag{9}$$

$$O^{2-}_{(ads)} + 1/2O_2 \leftrightarrow 2O^{-}_{(ads)} \tag{10}$$

$$2O^{-}_{(ads)} \leftrightarrow O_2^{-}_{(ads)} + e \tag{11}$$

$$O_2^{-}_{(ads)} \leftrightarrow O_{2(gas)} + e \tag{12}$$

$$NO_{(gas)} + O^{-}_{(ads)} \leftrightarrow NO_{2(gas)} + e \tag{13}$$

It is known that point defects on ZnO surfaces play an important role in gas sensor applications since they strongly influence electrical properties. For example, it has been demonstrated that ZnO with a high density of oxygen vacancies has high electrical conductivity [128,129]. Furthermore, it has been demonstrated that increasing the number of defects or oxygen vacancies on the surface of sensing materials considerably improves reactivity, thereby enhancing the sensing performance [129,130]. It was believed that oxygen vacancies could act as favorable adsorption sites for $NO_2$ molecules [131], leading to an increase in the electrostatic interaction of reactive $NO_2$ molecules with the ZnO NW surface and consequently resulting in an increase in gas sensitivity. Indeed, several theoretical results suggest that surface defects such as oxygen vacancies can control the chemical/electronic properties and adsorption characteristics of metal oxide surfaces [132,133]. Very recently, using density functional theory (DFT) calculations, An et al. [113] determined

that the adsorption energy of NO$_2$ on the oxygen vacancy site is $E_{ads}$ = −0.98 eV, which is three times larger than that on the perfect site ($E_{ads}$ = −0.30 eV), and hence, it was expected that the charge transfer from the oxygen vacancy site to the NO$_2$ adsorbate would be larger than that from the perfect site to the NO$_2$ adsorbate. This means that oxygen vacancies form stronger bonds with NO$_2$ molecules, attracting more charge from the ZnO surface than would occur with an oxygen-vacancy-free ZnO surface. This finding is consistent with the close correlation between the concentration of oxygen-vacancy-related defects and the sensitivity of ZnO-based NO$_2$ sensors [129,130]. Since NO$_2$ is a strong oxidizing gas with high electrophilicity (~2.28 eV), it directly chemisorbs on oxygen vacancy sites present on the sensor surface, such that the adsorbed NO$_2$ molecules can readily dissociate into O$^-$ adatoms and release NO gas, following the reaction shown in Equation (14) [134]. In such a case, one of the O atoms of NO$_2$ fills the oxygen vacancy site, and the weakly bonded NO can desorb from the surface [113]. In addition to this, the NO$_2$ adsorption can also occur on lattice oxygen (O$^{2-}_{(Lat)}$), which further contributes to the sensing performance.

$$NO_{2(ads)} + V_{\ddot{O}} \leftrightarrow (V_{\ddot{O}} - O^-{}_{(ads)}) + NO_{(gas)} \tag{14}$$

It is evident from the XPS and PL results (see Figures 5 and 6) that these ZnO NWs contained very few oxygen vacancies and very little deep interstitial oxygen on the surface, which may be considered as a factor contributing to the sensing performance. The more oxygen vacancies there are in the material, the more sensitive the sensor is. It should be noted here that the as-grown ZnO NWs possess excellent optical properties (i.e., a very high $I_{UV}/I_{DL}$ ratio); however, they can suffer from low sensitivity when the gas concentration is at the ppb level. Recently, much effort has been carried out to correlate the gas sensitivity with the point defects commonly present in ZnO [129,130,135–137].

In the gas sensors field, it has been indicated that the specific surface area is the predominant factor in determining the sensing performance of a material. Generally, the increase in the specific surface area of the material is favorable for increasing the number of the adsorbed target gas molecules (e.g., NO$_2$), thereby improving sensing performance. The specific surface area of the material is usually dependent on its morphological structure [138], which can be adjusted according to the method and conditions of preparation [40,41,139]. ZnO nanomaterials with a variety of sizes and shapes (including nanoparticles, nanorods, nanowires, nanotubes, nanofibers, nanoflowers, and hierarchical nanostructures) can be fabricated, and accordingly, their electrical and chemical properties as well as their specific surface area may be adjusted [139,140]. This is compatible with the gas-sensing characteristics of the material; the same materials with different morphologies can produce different gas-sensing properties [141–145]. It is anticipated that in addition to size/diameter [137] the length of nanowires can also have an impact on gas-sensing performance. For instance, Shooshtari et al. [146] investigated the effect of ZnO nanowire length on the sensing properties of CNT-ZnO samples. They indicated that as the lengths of the ZnO NWs increased from 0.5 to 1.5 um, the sensor response increased up to 30%, due to the adsorption site's effects. Furthermore, by increasing the aspect ratio of ZnO nanorods (length/diameter), the response to liquefied petroleum gas is greatly enhanced [31].

At present, the characteristic parameters of gas sensors, such as sensitivity, selectivity, stability, response time, recovery time, analyte concentration, working temperature, power consumption, and detection limit, have been reported many times in the literature [147–152]. Among them, selectivity is commonly used to evaluate the sensing performance of materials. A sensor is called selective if it only reacts with the target analyte and not with other components in a mixture. Hence, high selectivity confirms that the sensor gives precise information about the presence and concentration of gases. The selectivity of micro/nanostructured ZnO-based sensors can be calculated using Equation (15), as follows [150]:

$$K = \frac{S_{target}}{S_{interference}} \tag{15}$$

where $S_{target}$ and $S_{interference}$ are the responses of a sensor towards target gas and interfering gas, respectively. Chougule et al. [150] fabricated a gas sensor based on ZnO film for detecting $NO_2$, $H_2S$, and $NH_3$ and revealed high selectivity for $NO_2$ over $H_2S$ in comparison with $NH_3$. In another study, researchers selected seven different gases, including ethanol, toluene, methanol, $NH_3$, acetone, CO, and isopropanol, as target gases to investigate the gas response of a single-crystalline ZnO NW sensor [153]. They demonstrated that the ZnO NW sensor exhibits a higher response to ethanol vapor as compared with the other interference gases. The findings of their study indicated that the ZnO NW sensor had superior selectivity towards ethanol vapor. The selectivity of the sensor based on our ZnO NWs/Si-substrate samples is expected to be presented in another work (in preparation). The limit of detection (LOD) is considered an important measure that reflects the sensing performance. For high-performance sensor applications, a sensor should be able to detect extremely low concentrations of the gases. The lowest concentration of an analyte or gas that can be detected by a sensor under operating conditions is called its limit of detection. The LOD is calculated using the following Formula (16) [154]:

$$\text{LOD} = 3\frac{S_D}{\text{Slope}} \tag{16}$$

where $S_D$ is the intercept standard deviation, as given below (17):

$$\text{Standard deviation } (S_D) = \text{Standard error} \times \sqrt{N} \tag{17}$$

(here, N is the number of data points).

To calculate the LOD of the sensor, we plotted a graph of the sensitivity curve of the $NO_2$ concentrations. Figure 8 displays the linear fitting curve to $NO_2$ concentrations ranging from 2 to 10 ppm, indicating that the gas sensor has high linearity for $NO_2$ concentrations ranging from 2 to 10 ppm. The calculated slope and standard error values are 0.01625 and 0.00493, respectively. According to the definition of LOD, the value of LOD for $NO_2$ at 200 °C can be calculated using Equation (16):

$$\text{LOD} = 3\frac{S_D}{\text{Slope}} = 3\frac{0.00493 \times \sqrt{3}}{0.01625} = 1.57 \text{ ppm.}$$

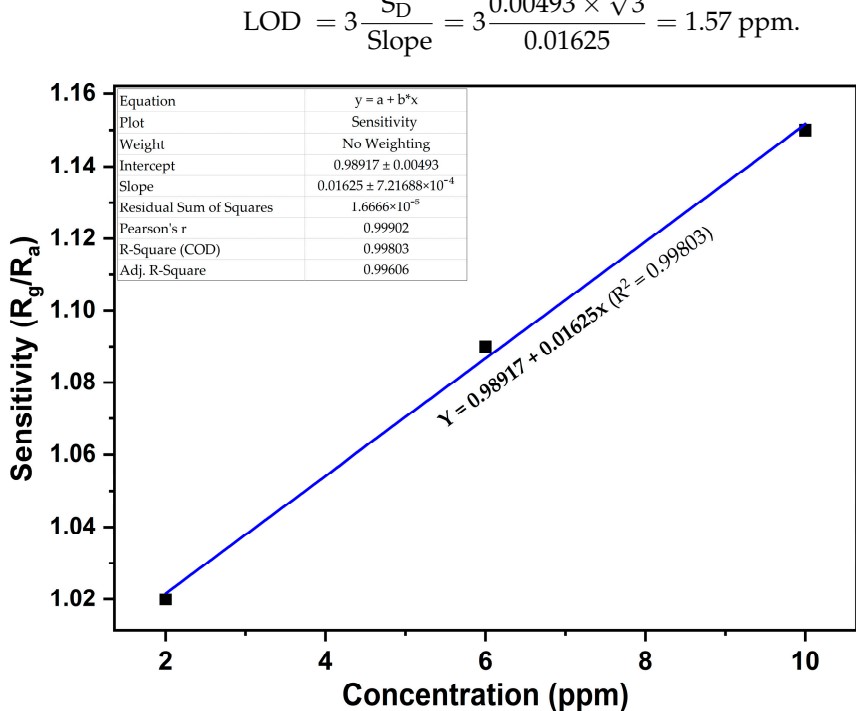

**Figure 8.** Sensitivity curve of ZnO NWs with different concentrations of $NO_2$ at 200 °C.

This result is comparable with previous findings for ZnO nanorod-based gas sensors, in which the lowest detection limits were estimated as 10 ppb at 250 °C [155], 50 ppb at 300 °C [156], and 100 ppb at 250 °C [151] for $NO_2$ gas.

## 4. Conclusions

In summary, well-aligned ZnO NWs were synthesized via the simple thermal evaporation route without using any catalyst or additive. The obtained samples were analyzed through various techniques such as XRD, FESEM, TEM, EDS, XPS, and photoluminescence spectroscopy. The FESEM and TEM images indicated that the as-prepared ZnO NWs are single-crystalline with a diameter in the range of 135–160 nm and an average length of roughly 3.5 μm, and grow vertically on the Si substrates with homogeneous distribution. The XRD and HRTEM results demonstrated that the ZnO NWs have a hexagonal wurtzite structure and are grown preferentially along the c-axis (0002). The XPS and PL results confirmed that the as-grown ZnO NWs are of high crystalline quality with very few oxygen defects on the surface. The room-temperature PL spectrum of the as-prepared ZnO NWs exhibited a sharp and strong UV emission (~380 nm) with a weak deep-level emission (~491 nm), indicating good optical properties. These properties make ZnO NWs attractive for application in fabricating high-performance optoelectronic devices. Additionally, the ZnO NWs were used as a sensor, which exhibited good sensing performance for $NO_2$ gas at an operating temperature of 200 °C; this was likely due to their high specific surface area and special atomic structure according to all the characteristics presented above.

**Author Contributions:** Conceptualization, T.V.K. and P.H.T.; methodology, T.V.K. and P.H.T.; software, T.V.K.; validation, T.V.K. and P.H.T.; formal analysis, P.H.T.; investigation, T.V.K.; resources, T.V.K. and P.H.T.; data curation, T.V.K.; writing—original draft preparation, T.V.K. and P.H.T.; writing—review and editing, T.V.K.; visualization, T.V.K. and P.H.T.; supervision, T.V.K.; project administration, T.V.K.; funding acquisition, T.V.K. All authors have read and agreed to the published version of the manuscript.

**Funding:** This research received no external funding.

**Data Availability Statement:** Not applicable.

**Acknowledgments:** We acknowledge the support of time and facilities from Ho Chi Minh City University of Technology (HCMUT), VNU-HCM for this study.

**Conflicts of Interest:** The authors declare no conflict of interest.

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
