# Peer review of "Thermal Evaporation Synthesis, Optical and Gas-Sensing Properties of ZnO Nanowires"

_crystals, doi:10.3390/cryst13091380_

Round 1
Reviewer 1 Report
This paper reported synthesis of vertically aligned single-crystalline ZnO nanowires (NWs) using the thermal evaporation method without any catalyst or additive. Characterization techniques confirmed uniform distribution, hexagonal wurtzite structure, and favorable optical properties. The ZnO NWs demonstrated excellent sensing performance for NO2 gas at 200°C, attributed to their high specific surface area and overall quality. This article has many laboratory data and can be considered for publication after considering the following comments:
1- Consider mentioning the novelty or contribution of the study in the abstract.
2- Consider including a brief mention of the specific results or findings obtained of gas sensing results from the study in the abstract.
3- The number of reference articles is very large and several articles are referenced for a scientific proposition. While it's important to support your statements with references, the excessive use of citation numbers makes the text appear cluttered. Consider integrating the citations more smoothly into the narrative or consolidating multiple references when possible.
Introduction:
4- Clarify why ZnO is a suitable material for gas sensing and mention its unique properties that contribute to its effectiveness.
5- Include a clear statement of the research objective or aim of the study.
6- When mentioning the different morphologies of ZnO nanostructures, briefly explain their advantages and limitations.
Materials and Methods
7- The details of the material used (supplier, etc.) must be stated.
Results and Discussion
8- How (002) relates to the growth mechanism or gas sensing performance of the ZnO NWs?
9- It would be helpful to discuss the significance of (IUV/IVis) ratio and how it relates to the optical properties and potential applications of the NWs.
10- How was the peak fitting performed, and what criteria were used to determine the number and positions of the fitted peaks in PL?
11- It would be helpful to elaborate on the importance of low DL defect concentration in terms of the NWs' optical properties and potential applications.
12- Mention the effect of the ZnO nanowire length on sensing properties (refer to DOI: 10.3390/chemosensors10060205).
13- Provide a more in-depth discussion of the gas sensing properties of the ZnO NWs for NO2, including the detection limit.
It would be beneficial to discuss the selectivity of the ZnO NWs as gas sensors for NO2 and how they distinguish NO2 from other gases that may be present in the environment.
The English writing of the text is generally clear and understandable. However, there are a few areas where improvements can be made for clarity and precision. Some sentences are quite long and complex, which can make them difficult to follow.
Author Response
Dear reviewers:
Thank you so much for your careful evaluation of our manuscript and the insightful comments. Those comments are all valuable and very helpful for revising and improving our paper, as well as the important guiding significance to our researches. We have studied comments carefully and have made correction which we hope meet with approval. Revised portion are marked with red in the paper.
We tried our best to improve the manuscript and made some changes in the manuscript. These changes will not influence the content and framework of the paper. We appreciate for Reviewers’ warm work earnestly, and hope that the correction will meet with approval.
Once again, thank you very much for your comments and suggestions.
Sincerely yours,
Tran Van Khai

Reviewer 2 Report
In this manuscript ZnO nanowires were synthesized with thermal evaporation method, then structural, optical and NO2 gas sensing properties of these nanowires were investigated. The authors previously published ZnO nanowire fabrication and characterization in details (Van Khai, Tran, et al. "Diameter‐and density‐controlled synthesis of well‐aligned ZnO nanowire arrays and their properties using a thermal evaporation technique." physica status solidi (a) 209.8 (2012): 1498-1510.). They have to clarify what is new in the manuscript within the scope of the synthesis and characterization compared to their previous paper. The last part NO2 sensing properties of ZnO nanowire is not enough in the current form.
Author Response
Dear reviewer:
Thank you so much for your careful evaluation of our manuscript and the insightful comments. Those comments are all valuable and very helpful for revising and improving our paper, as well as the important guiding significance to our researches. We have studied comments carefully and have made correction which we hope meet with approval. Revised portion are marked with red in the paper.
We tried our best to improve the manuscript and made some changes in the manuscript. These changes will not influence the content and framework of the paper. We appreciate for Reviewers’ warm work earnestly, and hope that the correction will meet with approval.
Once again, thank you very much for your comments and suggestions.
Sincerely yours,
Tran Van Khai

Round 2
Reviewer 2 Report
In this manuscript ZnO nanowires were synthesized with thermal evaporation method, then structural, optical and NO2 gas sensing properties of these nanowires were investigated. Following revisions are suggested
11. In abstract:
a) “…highly oriented along the [001] direction”
If this is the preferred surface, the value should be (0001). If this is the bragg surfaces obtained in XRD, it should be (002). Please revise also in results and discussion
b) “detected NO2 gas at very low concentrations, as low as ~2 ppm”
For NO2 gas detection 2 ppm is very high according to [Kuklinska, K., Wolska, L., & Namiesnik, J. (2015). Air quality policy in the US and the EU–a review. Atmospheric Pollution Research, 6(1), 129-137.], Please revise
22. In Materials and methods
a) “The vertical distance between the zinc source and the substrate was about 1.5–5 mm, with a downstream separation of 5–25 mm”
The position of Zn source and substrate is critical as you previously observed [ref 59]. The experimental conditions must be clearly defined for obtained ZnO nanowire that used in sensor device.
b) “The zinc source was thermally vaporized to synthesize ZnO NWs at atmospheric pressure under Ar at a flow rate of 150 mL min -1 for 90 min at 620 ◦C.”
This sentence is not required, previous sentence gives same information.
c) How do you obtain 2 – 10 ppm NO2 gas mixture? What is the concentration of source (NO2 gas tube)? The IDT electrode dimensions is very important. Also the type of Si is crucial. Please clarify the experimental section
33. In Results and Discussion:
a) As you mentined Si-ZnO nanowire is less studied for NO2 sensor application, moderate temperature (200 C) could not be used. You have to measure NO2 sensing at various temperature and then decided the optimum working temperature.
b) Your device is semiconductor (Si) – semiconductor (ZnO) heterostructure. So, you have to mention NO2 sensing mechanism in this direction also.
c) Please define sensitivity that is important parameter of a sensor.
d) You mentioned about selectivity in page 15, but there is not any results. Do you measure any different gases for selectivity?
44. For whole paper;
The similarity value obtained from Turnitin is very high (55% without references).
Please reduce it
Author Response
Dear Reviewer,
First, we thank the reviewer for careful reading of the paper and your very helpful comments. We performed the suggested corrections and improved the relevant sections of the manuscript to address the questions and comments point-by-point. Changes were highlighted throughout the manuscript. So, we hope our responses have the potential to match the questions of the reviewer.
Thanks again for taking the time to read and write in – I have learned a lot from your comments, questions and suggestions.
Sincerely yours,
Tran Van Khai
